# Improving the Storage Stability of Soy Protein Isolate through Annealing

**DOI:** 10.3390/foods13040615

**Published:** 2024-02-18

**Authors:** Shenzhong Zou, Zhaojun Wang, Maomao Zeng, Zhiyong He, Jie Chen

**Affiliations:** 1School of Food Science and Technology, Jiangnan University, Wuxi 214122, China; 6200113243@stu.jiangnan.edu.cn (S.Z.); 8201911201@jiangnan.edu.cn (Z.W.); zyhe@jiangnan.edu.cn (Z.H.); 2State Key Laboratory of Food Science and Resources, Jiangnan University, Wuxi 214122, China

**Keywords:** annealing, soy protein isolate, solubility, intrinsic fluorescence spectrum

## Abstract

This study investigated the effect of annealing treatment on the stability of soy protein isolate (SPI) during storage. Different SPI samples with varying denaturation levels were subjected to varying annealing temperatures and durations before being stored at 37 °C for 12 weeks to assess their stability. Our findings revealed that annealing at 65 °C for 30 min significantly mitigated protein deterioration, improving the stability of highly denatured proteins during storage. Surface hydrophobicity and endogenous fluorescence analyses indicated that this annealing condition induced protein structure unfolding, an initial increase in SPI hydrophobicity, and a blue shift in the maximum absorption wavelength (λ_max_). The slowest increase in hydrophobicity occurred during storage, along with a red shift in the maximum absorption wavelength by the 12th week. These results suggest that annealing treatment holds promise for mitigating the issue of reduced SPI stability during storage.

## 1. Introduction

Soy protein is widely used in the food industry due to its versatility, providing rich nutrition for vegetarians and serving as a key ingredient in various products like soy milk, tofu, sausage [1,2]. However, a persistent issue in the soy protein industry worldwide is the reduction in protein quality during storage and transportation, notably evident in its reduced solubility [3]. For instance, the solubility of SPI in water (2%) decreased by 63% after one year of storage at 42 °C [4]. Solubility is crucial for the functionality of soy proteins, including emulsification, gelation and foaming [5]. Regrettably, current research has not fully addressed the factors leading to solubility loss during storage, and effective strategies to enhance solubility under these conditions are lacking.

Researchers have explored the problem of reduced soy protein solubility during storage. Their investigations have highlighted three primary factors contributing to the this issue: thermal aggregation during processing and storage, oxidation of fats and oils, and chemical reactions such as the Maillard reaction [6]. Efforts to mitigate these factors have included storing proteins at low temperatures (4 °C) to reduce thermal aggregation, adding antioxidants, or employing vacuum packaging to hinder protein oxidation [6,7]. However, these approaches often entail high costs and yield only modest improvements.

Soy protein can undergo inactivation or aggregation due to internal stress during production, processing and storage, influenced by factors like temperature and pH [8]. For example, in the preparation of SPI, raw soybeans undergo a series of procedures, including grinding, degreasing, alkaline solubilization, acid precipitation, spray-drying and sterilization [9,10]. These processes inevitably lead to varying degrees of internal stress, resulting in protein aggregation and compromising initial stability and excellent functionality, which eventually results in reduced solubility during storage [6]. Therefore, we introduce the concept of annealing, which could potentially improve the storage stability by releasing internal stress.

Annealing, originally applied to metal, involves heating a material to high temperatures and gradually cooling it to reorganize its internal structure, relieve internal stress, and improve steel grain size and distribution [11]. This controlled heating process can give materials a more uniform and favorable microstructure, enhancing material toughness and flexibility [12,13]. Remarkably, annealing has extended its applications beyond metal and glass, proving successful in other polymer substances and biological macromolecules. Polymers like polyethylene, polypropylene, polylactic acid can undergo annealing to enhance grain size and concentration. In the realm of polymer films, annealing reduces stress, improves flatness and enhances stability [14]. In the context of biological macromolecules, such as starch and ribosomes, annealing has been found to change molecular chain arrangements and structures, thereby improving physical properties, stability and functionality [15,16].

However, little has been reported on the annealing treatment of soybean protein isolate. The objective of this study was to improve SPI stability during storage through annealing. Prior to spray-drying, we prepared proteins with varying degrees of denaturation by subjecting them to heat treatment. This yielded three groups: highly denatured samples exposed to ultra-high-temperature sterilization (135 °C, 5 s); lowly denatured samples without sterilization; and native samples without thermal denaturation, directly freeze-dried. The low-denaturation and high-denaturation samples were subsequently annealed at different temperatures (45, 55, 65, and 75 °C) for various durations (0, 5, 10, 30, 60, and 120 min). These samples were then stored at 37 °C for 12 weeks to observe and analyze changes in solubility.

## 2. Materials and Methods

### 2.1. Materials

Soybean (Golden Soybean 626, harvested in 2020) was purchased from Fengyuan Seed Co. Lianyungang, China. The soybean contained 40.52% ± 1.72% protein, 19.57% ± 1.68% lipid, 12.52% ± 0.89% moisture, and 4.40% ± 0.53% ash. Analytical-grade chemicals were purchased from Sigma Aldrich Trading Co Ltd. (Shanghai, China). Deionized water was used for all reagents and sample preparations.

### 2.2. Preparation of Soy Protein Isolate

Soy protein isolate (SPI) was prepared from soybean employing an alkaline pH extraction-isoelectric precipitation method, as described by Fu et al. (2023) [17]. Following neutralization to pH 7.0, one set of samples was subjected to vacuum freeze-drying and designated the native sample (NSPI). Another set of samples underwent a spray-drying process, designated the lowly denatured sample (LSPI). Additionally, samples exposed to ultra-high-temperature transient sterilization at 135 °C for 5 s were labeled the high-denaturation samples (HSPI). The protein content of the extracted SPI exceeded 90% (dry basis), with a nitrogen conversion factor of 6.25.

### 2.3. Annealing Treatments

High-denaturation and low-denaturation samples were uniformly dispersed on the surface of an iron basin, and subjected to annealing treatment upon reaching the target temperature. Annealing temperatures spanned four gradients: 45, 55, 65, 75 °C. Annealing durations encompassed six intervals: 0, 5, 10, 30, 60, 120 min. Following annealing, the samples were vacuum-sealed and stored at 37 °C in the absence of light for 12 weeks. Subsequently, samples were retrieved in the 1st, 2nd, 6th, 10th, and 12th weeks for assessment.

### 2.4. Solubility

A precise mass of SPI was weighed and dispersed in deionized water (1%, *w*/*v*). The mixture was stirred at room temperature for 1.5 h to ensure complete dissolution. After standing for 2 min, the upper layer of protein solution was separated, followed by centrifugation for 15 min at 10,000× *g*. Protein content was determined using the Kjeldahl nitrogen determination method.
(1)$$\mathrm{Solubility}=\frac{\mathrm{supernatant}\;\mathrm{protein}\;\mathrm{concentration}}{\mathrm{sample}\;\mathrm{protein}\;\mathrm{concentration}}$$

### 2.5. Particle Size Distribution

The particle size distribution of SPI samples was determined using a Zetasizer Nano-ZS instrument (Malvern Instruments, Worcestershire, UK). SPI samples were dissolved in deionized water and prepared as a 0.1% solution, and the refractive index and absorption parameters of the SPI sample were 1.450 and 0.001, respectively.

### 2.6. Molecular Weight Distribution

A 1% SPI solution was prepared and passed through a 0.45 μm aqueous membrane with a syringe. Gel permeation chromatography was performed using a high-performance liquid chromatography system (Shimadzu, Kyoto, Japan), featuring a KW-804 protein gel column and an ultraviolet detector. The mobile phase comprised 50 mmol/L pH 7.0 phosphate buffer, 0.3 M ionic strength, and an elution rate of 1.0 mL/min and was detected at 280 nm.

### 2.7. Surface Free and Total Free Sulfhydryl Groups

Total free and surface free sulfhydryl groups were determined using 5,5′-dithiobis-(2-nitrobenzoic acid; DTNB) according to the methods reported by Beveridge et al. (1974) and Tang et al. (2009) with slight modifications [18,19]. A 2 mg/mL SPI sample (1 mL) was mixed with 4 mL of Tris-glycine buffer (0.086 M Tris, 0.09 M glycine, 4 mM Na2-EDTA, pH 8.0). Subsequently, 0.04 mL of DNTB solution was introduced, and the mixture was incubated for 15 min at room temperature in the dark. Absorbance at 412 nm was recorded for surface sulfhydryl groups’ determination. Tri-glycine buffer containing 8 M urea was prepared for total free sulfhydryl groups’ determination. Then, 1 mL of protein solution was mixed with 4 mL of urea-guanidinium hydrochloride solution, followed by the addition of 0.04 mL of DNTB solution. Incubation at room temperature in the dark for 15 min was followed by absorbance measurement at 412 nm to calculate total free sulfhydryl groups using the following formula:(2)$$\mathrm{SH}\left(\frac{\mu \mathrm{mol}}{g}\right)=75.53\times \frac{A_{412}}{C}$$

SH denotes the sulfhydryl group, A_412_ denotes the absorbance of the sample at a wavelength of 412, and C denotes the concentration of protein.

### 2.8. Sulfhydryl and Disulfide Content

Disulfide bonds were quantified according to the method reported by Hu, Hao et al. (2013) with slight variations [20]. Then, 1 mL protein solution was mixed with 0.05 mL of β-mercaptoethanol and 4 mL of urea-guanidinium hydrochloride (8 mol/L urea and 5 mol/L guanidinium hydrochloride) solution. The mixture was incubated for 1 h at 25 °C, after which 10 mL of 12% TCA (trichloroacetic acid) was added. Further incubation in a water bath at 25 °C for 1 h was followed by centrifugation at 10,000× *g* for 10 min. The precipitate was dispersed in 5 mL of 12% TCA, and then centrifuged to remove β-mercaptoethanol. The precipitate was dispersed in 5 mL of 12% TCA, and β-mercaptoethanol was removed through centrifugation. This step was repeated twice. Finally, the precipitate was dissolved in 2.5 mL of a Tris-Gly solution of urea-guanidine hydrochloride, and 0.04 mL of DNTB solution was added. Absorbance was measured at 412 nm. Disulfide bonds content was calculated using the following formula:(3)$$-S-S-=\frac{{\mathrm{SH}}_{T}{-\mathrm{SH}}_{F}}{2}$$

-S-S-denotes the disulfide bond, SH_T_ denotes the total sulfhydryl and SH_F_ denotes the total free sulfhydryl.

### 2.9. Surface Hydrophobicity

The surface hydrophobicity of proteins was evaluated using a 1-anilino-8-naphthalenesulfonic acid (ANS) fluorescent probe, with slight modifications, following the method of Xu et al. (2016) [21]. SPI was diluted within a concentration range of 0.5–2.5 mg/mL using a 0.01 mol/L phosphate buffer at pH 7.0. Separately, 5 mL of sample solution with different concentrations were prepared, and 40 μL of 8 mM ANS was added and fully shaken. Fluorescence intensity was measured using a 650-60 fluorescence spectrophotometer (Hitachi, Tokyo, Japan), employing an excitation wavelength of 365 nm, an emission wavelength of 520 nm, and a slit correction of 5 nm. A fluorescence intensity versus protein concentration curve was constructed, and the slope of the regression equation provided the surface hydrophobicity value of the protein.

### 2.10. Intrinsic Fluorescence Spectroscopy

Fluorescence spectra of SPI were separately determined by the method of reference in the work of Zhang et al. (2016) using a 650-60 fluorescence spectrophotometer (Hitachi, Tokyo, Japan) with slight modifications [22]. SPI was dissolved in 10 mM phosphate buffer at pH 7.0 to achieve a final protein concentration of 0.05 mg/mL. The scanning emission wavelength ranged from 290 to 460 nm, with an excitation wavelength of 280 nm. Both excitation and emission slits were set to 5 nm, and the scanning rate was established at 1200 nm/min, with a voltage of 600 V.

### 2.11. SDS-PAGE Analysis

Electrophoretic patterns of SPI were visualized by SDS-PAGE, conducted under both reducing (with the addition of β-mercaptoethanol) and non-reducing (without β-mercaptoethanol) conditions. SPI samples were configured at a concentration of 2 mg/mL, and each SPI sample and molecular weight markers were loaded in volumes of 10 μL. A 4% concentrated gel and 12% separated gel were employed for analysis. Gel electrophoresis was performed using a Bio-Rad Mini Protein Electrophoresis System (Bio-Rad Laboratories, Hercules, CA, USA) at 40 V for the concentrated gel and at 80 V for the separated gel. To achieve a clear background, the gel was washed three times with boiling water, stained with rapid staining solution for 30 min, and then decolorized with deionized water for 120 min.

### 2.12. Statistical Analysis

This experimental dataset included two replications in three parallels. Statistical analysis was carried out using Excel 2010 and SPSS 25 software to assess potential significant differences among the results.

## 3. Results and Discussion

### 3.1. Solubility

Protein solubility was continuously tracked throughout storage to assess the solubility of SPI (Figure 1 and Figure A1). The solubility of all samples showed a decreasing trend with increasing storage time, which is consistent with previous findings [4]. However, samples with different degrees of denaturation demonstrated distinct solubility reductions during storage. NSPI displayed the smallest decline in solubility, followed by LSPI, while HSPI exhibited the most significant decrease. This variation can be attributed to the differing degrees of initial protein aggregation, where a higher degree of initial aggregation correlated with poorer protein stability during storage [6]. In addition, different annealing treatments had a significant effect on the change in storage solubility for HSPI, but not for LSP, probably due to the less severe structural damage in LSPI, rendering annealing treatments less influential.

The effects of different annealing conditions on solubility change during HSPI storage were diverse. Within the annealing temperature range of 45 to 55 °C, there were no significant differences in protein solubility during storage. This observation could be attributed to the relatively low annealing temperature, which did not fully unfold the internal structure or eliminate residual internal stresses from processing. Conversely, an annealing temperature of 75 °C accelerated protein deterioration, probably because the elevated temperature caused significant protein aggregation. The most effective annealing temperature for maintaining protein storage solubility proved to be 65 °C. Despite including initial protein aggregation, a 30 min annealing period for HSPI resulted in the slowest solubility decline, maintaining an 80% solubility even by the 12th week. These findings underscore the potential of suitable annealing conditions in enhancing protein stability during storage.

### 3.2. Particle Size Distribution

Since the annealing treatment at 65 °C had a significant effect on the improvement of solubility during protein storage, the particle size distribution of all annealed samples at this temperature was traced and measured. Particle size distribution serves as a vital parameter for evaluating the extent of protein aggregation and plays a crucial role in charactering protein stability [23]. Figure 2 illustrates the particle size distributions of NSPI and HSPI annealed at 65 °C over a 12-week storage period. Their distributions manifested a distinct bimodal pattern, wherein HSPI, having undergone thermal denaturation, exhibited protein particle aggregation, resulting in a rightward shift (indicating larger particle sizes) of the two peaks [24]. At week 0, the protein particle size increased with increasing annealing time (Figure 2a), probably attributed to the aggregation induced by the heat treatment. The particle sizes of all protein samples showed an increasing trend with increasing storage time. The particle size of HSPI displayed different degrees of increase over different annealing durations, potentially linked to the aggregation kinetics of the protein [6]. Notably, HSPI subjected to a 30 min annealing displayed smaller particle sizes throughout the storage period, although accelerated storage resulted in an increase in particle size for this sample, reaching a minimum in the 10th week and maintaining a similar trend until the 12th week (Figure 2d,e). This suggested that brief annealing at 65 °C (5 and 10 min) did not yield an improved long-term storage outcome; instead, it may have induced in internal stress, leading to the increased particle size. However, excessively prolonged annealing durations (60 and 120 min) may have triggered a series of reactions resulting in protein aggregation and subsequent particle size enlargement.

### 3.3. Molecular Weight Distribution

To further illustrate the effect of annealing treatment on HSPI stability throughout storage, we analyzed the relative molecular mass distribution of the proteins, as shown in Figure 3 and Figure A2. To facilitate comparison among samples, we categorized into four segments based on relative molecular mass interval as represented by group A, B, C and D. Interval A (>1000 kDa) corresponds to large-size aggregates, interval B (670–1000 kDa) to medium-size aggregates, interval C (43–670 kDa) to components of 11 s and 7 s (340 and 200 kDa, respectively) globular proteins, and interval D (<43 kDa) to dissociated small-size portions of the corresponding proteins. NSPI exhibited the lowest percentage of group A and the highest percentage of group D, whereas all HSPI samples exhibited a higher percentage of group A, which could be attributed to protein denaturation and aggregation [25]. At week 0, different annealing treatments resulted in increased percentage of group A, probably due to heat-induced aggregation and the formation of disulfide bonds, resulting in increased macromolecular content and reduced solubility [26]. With the increase in storage time occurring at 37 °C, the content of group A in all samples displayed an upward trajectory, indicating accelerated protein aggregation [6]. However, it is worth noting that the macromolecular content of HSPI did not witness significant numerical growth, probably due to the filtration of insoluble aggregates formed by the samples after the sixth week. In addition, the duration of annealing exerted a notable influence on the molecular weight distribution. In particular, a 30 min annealing time corresponded to lower protein macromolecule content, reaching its lowest by the 12th week. This underscores the pivotal role of annealing duration in preserving protein structure [15].

### 3.4. Sulfhydryl Groups Contents

Sulfhydryl groups (-SH) and disulfide bonds (-S-S-) within soy proteins can be interconverted under specific conditions, significantly influencing protein functional properties [27]. Figure 4 presents changes in surface sulfhydryl groups, total free sulfhydryl groups, total sulfhydryl groups and disulfide bonds of the protein during storage. HSPI exhibited higher disulfide bond and total free sulfhydryl group contents compared to NSPI. This difference may elucidate why thermal denaturation treatment can lead to the aggregation of free sulfhydryl groups to form disulfide bonds, consequently explaining higher solubility of NSPI [28].

At week 0, there was a small increase in disulfide bond content with prolonging annealing time, indicating that the annealing induced minor protein aggregation, which is consistent with the molecular weight and particle size distribution results (Figure 2 and Figure 3). As storage duration extended, both surface sulfhydryl groups and total sulfhydryl groups showed a decreasing trend. However, the extent of this change remained relatively consistent across different samples, suggesting that annealing had a minimal effect on the change in surface sulfhydryl groups and total sulfhydryl groups. Nevertheless, different annealing treatments affected the variations in total free sulfhydryl groups and disulfide bonds during protein storage. Proteins annealed for 30 min displayed the slowest rate of increase in disulfide bonds. In weeks 10 and 12, the quantity of disulfide bonds in HSPI samples subjected to 30 min annealing time was lower than that in all other samples. This, in turn, contributed to their relatively high solubility at the end of the storage period. Consequently, annealing treatments reduced protein aggregation rates during storage, enhancing overall stability. In addition, the protein samples exhibited a slightly swifter decline in total free sulfhydryl groups compared to the rate of disulfide bond formation, probably due to the conversion of free sulfhydryl groups into other sulfur-containing substances [29].

### 3.5. Surface Hydrophobicity

The protein surface hydrophobicity provides insight into its degree of denaturation and the interactions among protein molecules [30]. ANS fluorescence data were employed to probe structural changes during protein storage (Figure 5). The results reveal that HSPI exhibited higher hydrophobic values compared to NSPI, probably due to the denaturation treatment that partially unfolded the protein structure, exposing previously buried hydrophobic regions within the protein molecule [31].

At week 0, the protein hydrophobicity increased with extended annealing time. Prolonged heat treatment appears to promote a greater unfolding of the protein structure, leading to more substantial exposure of hydrophobic groups and a more pronounced hydrophobic effect, resulting in reduced initial protein solubility [32]. As storage time progressed, proteins subjected to different annealing treatments exhibited varied degrees of increased hydrophobicity. Notably, the 30 min annealing treatment maintained the lowest hydrophobic value throughout the storage period, despite initially elevating protein hydrophobicity. This phenomenon may stem from the fact that the annealing leads to varying degrees of protein unfolding and exposure of hydrophobic groups. However, suitable annealing conditions help remove most of the stress associated with the production process, promoting a more stable structure within and between protein molecules. This limits the exposure of hydrophobic groups during storage, resulting in a relatively modest increase in hydrophobicity [33]. As a result, the hydrophobic value showed a relatively minor increase. Moreover, this enhanced stability of protein structure during storage can be associated with a reduced level of aggregation. From a microscopic perspective, it can be deduced that annealing contributed to the improved stability of the protein throughout storage.

### 3.6. Intrinsic Fluorescence Spectrum

Endogenous fluorescence spectroscopy is a valuable tool for tracking changes in the polarity of the microenvironment surrounding aromatic amino acids (tyrosine, tryptophan, and phenylalanine). This approach provides a direct characterization of the tertiary conformational changes experienced by proteins [34]. Typically, proteins exhibit fluorescence in the range of 320–350 nm, which is especially notable in the case of tryptophan residues following excitation at 290 nm [35]. The inherent fluorescence spectra during protein storage are displayed in Figure 6. Throughout the storage period, NSPI consistently exhibited the lowest fluorescence intensity value (FI) compared to HSPI. This difference can be attributed to the denaturation treatment, which likely led to protein unfolding, exposure of tryptophan residues, and an increased presence of chromogenic sites, resulting in elevated FI [36].

At week 0, the annealing treatments induced higher FI of the protein. FI gradually increased with extended annealing time, accompanied by a blue shift in the maximum absorption wavelength (λ_max_), shifting from 337.5 nm to 334.5 nm. This shift may be due to the annealing-induced unfolding of the protein structure, altering the tryptophan microenvironment from relatively hydrophilic to relatively hydrophobic [37]. As storage time progressed, the 30 min annealed proteins displayed a gradually decrease in FI, reaching a minimum after week 10 compared to other annealed samples. λ_max_ shifted from 334 nm to 337 nm, compared to the unannealed protein, by week 12. This phenomenon suggested that annealing may have alleviated most of the residual stresses associated with protein processing. This led to the formation of a relatively stable protein structure and a deceleration of aggregation. Thus, the fluorescence intensity from aromatic amino acid residues decreased, and the tryptophan microenvironment transitioned from relatively hydrophobic to relatively hydrophilic [38]. Therefore, annealing appears to facilitate the formation of a stable protein conformation, mitigate hydrophobic protein aggregation, and reduce fluorescence coloration sites, aligning well with the solubility and surface hydrophobicity.

### 3.7. SDS–PAGE

Figure 7 and Figure A3 shows the SDS-PAGE patterns of SPI under different annealing conditions. Under reduced SDS-PAGE conditions, SDS disrupted non-covalent bonds between aggregates and disrupted S-S bonds due to the presence of SDS and β-mercaptoethanol [39]. In contrast, under non-reducing SDS-PAGE conditions, S-S bonds remain intact, while SDS solely denatured non-covalent bonds between aggregates. Under both reducing and non-reducing conditions, all samples showed major components of SPI, namely the 7S and 11S protein subunits [39]. The 7S fraction consists of three subunits, α, α, and β; the 11S component is a hexamer consisting of an acidic subunit A and a basic subunit B, connected by S-S bonds [40]. No significant differences in subunit content and composition were observed between the protein samples at week 0 and week 12 under reducing SDS-PAGE conditions, indicating that the annealing and accelerated storage treatments did not alter the protein composition. Under non-reducing SDS-PAGE conditions, the HSPI samples exhibited an increase in the content of macromolecular bands but a significant decrease in the content of AB subunits compared to NSPI. This suggests that soluble aggregates, potentially generated during heat treatment, contain A and B subunits linked by disulfide bonds. The relatively high solubility of the sample that underwent a 65 °C, 30 min treatment during storage could be attributed to the soluble aggregates.

## 4. Conclusions

Annealing treatment improved the stability of HSPI during storage, despite the initial occurrence of protein aggregation. The 65 °C, 30 min annealing exhibited the most favorable outcomes, displaying the slowest decline in protein solubility, the least protein aggregation, smaller particle size, lower large molecular weight content and a more gradually formation of disulfide bond. The annealing treatment induced changes in the tertiary structure of the protein. In particular, the 65 °C, 30 min annealing resulted in a slower increase in protein hydrophobicity, a gradual decrease in FI, and an enlargement of λ_max_. This is likely due to the fact that the annealing process eliminated the residual stress of the protein processing, thereby maintaining a relatively stable structure. In conclusion, annealing treatment provides a promising approach to mitigating decreases in the solubility of soy protein during its storage. Future investigations should explore the combination of annealing with chemical methods to further enhance the stability of soy protein during storage.

## Figures and Tables

**Figure 1 foods-13-00615-f001:**
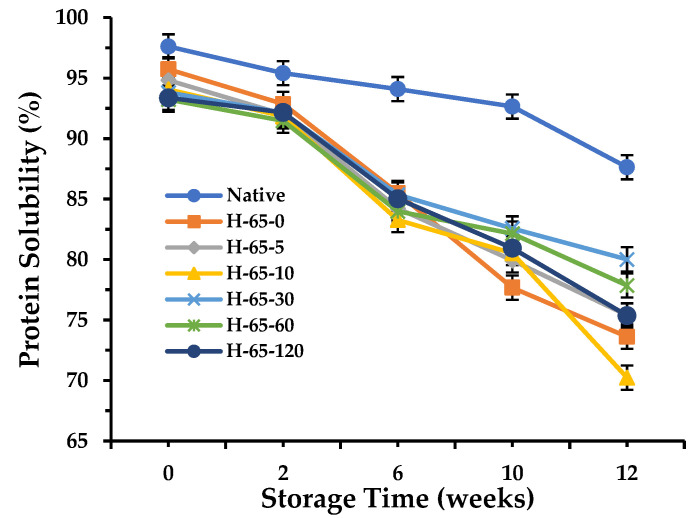
The solubility of soy protein isolate with or without annealing treatment as a function of storage durations. NSPI: native soy isolate protein; HSPI: denatured soy isolate protein; H-65-5 denotes highly denatured soy protein isolate subjected to annealing treatment at 65 °C for 5 min; H-65-30 denotes highly denatured soybean isolate annealed at 65 °C for 30 min.

**Figure 2 foods-13-00615-f002:**
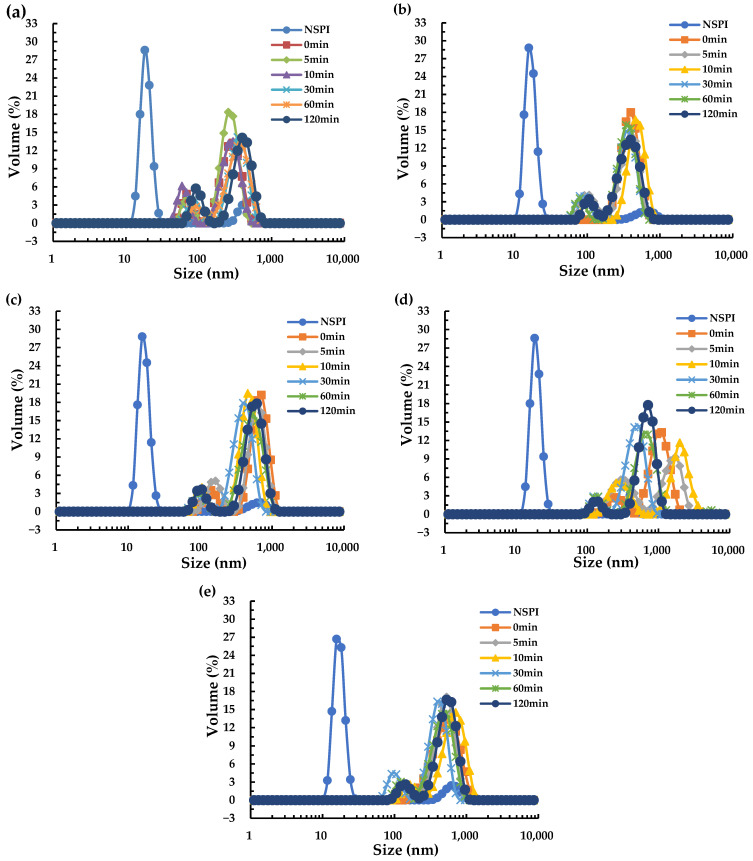
Particle size distribution of HSPI at an annealing temperature of 65 °C, different annealing times, and different storage times; (**a**–**e**) indicate the particle size distribution of the samples at 0, 2, 6, 10, and 12 weeks, respectively.

**Figure 3 foods-13-00615-f003:**
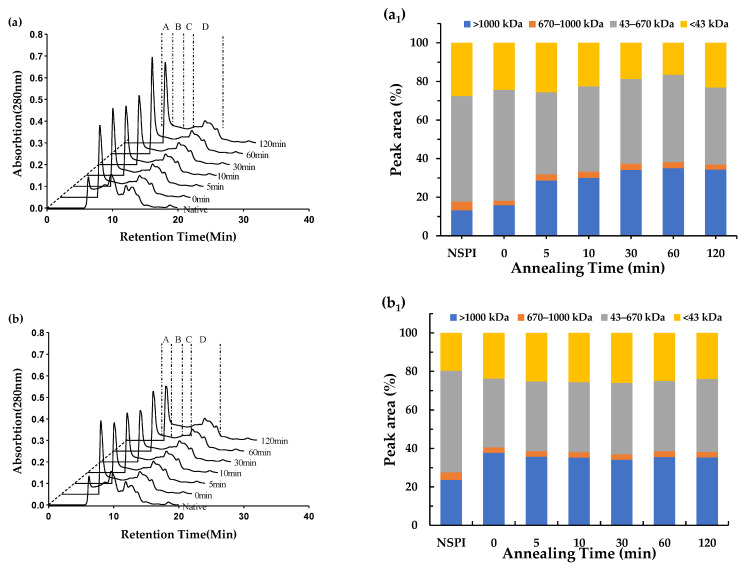
Molecular weight distribution of HSPI samples with an annealing temperature of 65 °C, different annealing times, and different storage times; (**a**,**a_1_**,**b**,**b_1_**) indicate the molecular weight distribution of the samples at 0 and 10 weeks, respectively. Intervals A, B, C and D indicate molecular weights greater than 1000 kDa, 670 kDa to 1000 kDa, 43 kDa to 670 kDa and less than 43 kDa, respectively.

**Figure 4 foods-13-00615-f004:**
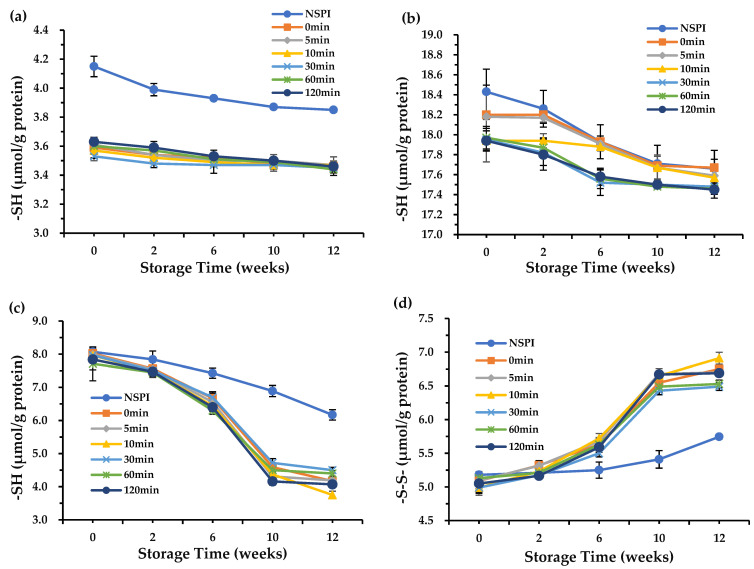
Sulfhydryl group content of HSPI samples with an annealing temperature of 65 °C, different annealing times, and different storage times; (**a**–**d**) indicate changes in surface sulfhydryl groups, total sulfhydryl groups, total free sulfhydryl groups, and disulfide bonds in the samples during the storage.

**Figure 5 foods-13-00615-f005:**
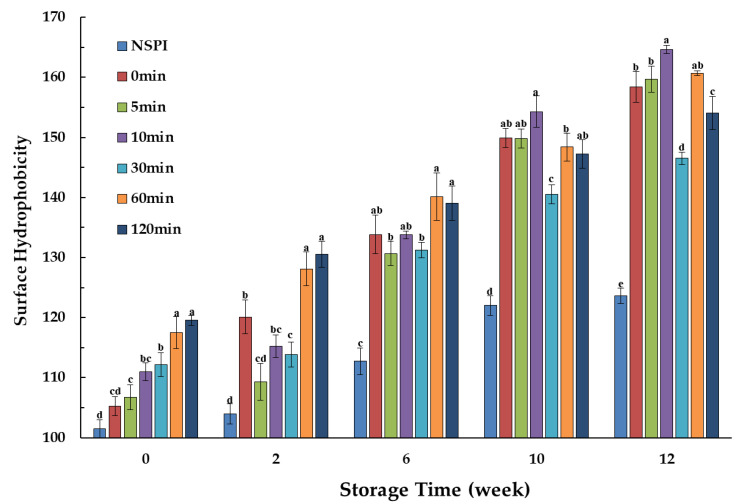
Surface hydrophobicity of HSPI samples at an annealing temperature of 65 °C, different annealing times, and different storage times. Different letters indicate significant differences between different samples in the same week (*p* < 0.05).

**Figure 6 foods-13-00615-f006:**
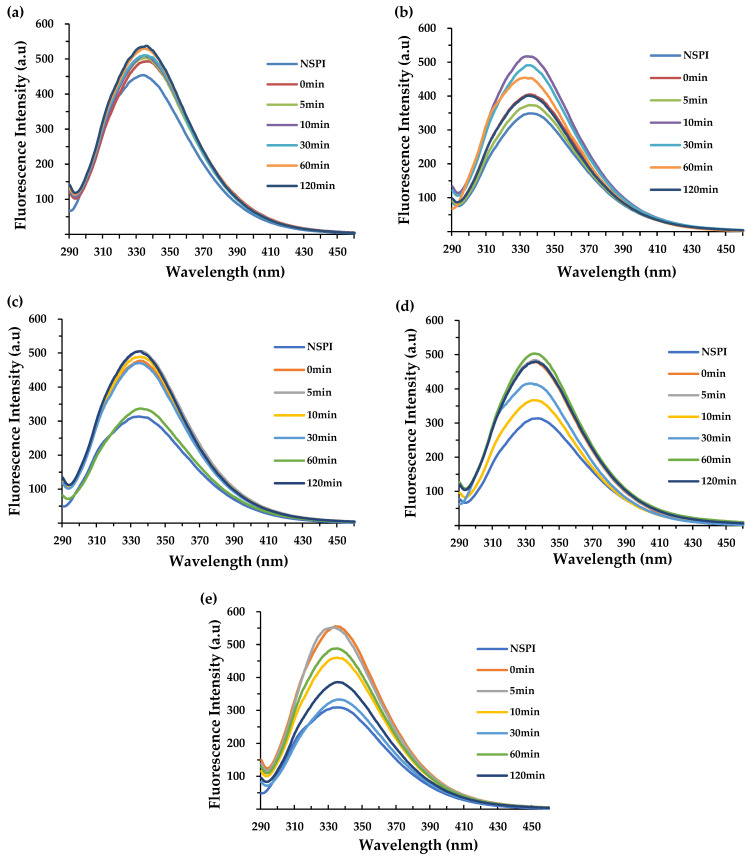
Intrinsic emission fluorescence spectra of HSPI samples with an annealing temperature of 65 °C, different annealing times, and different storage times. (**a**–**e**) Intrinsic emission fluorescence spectra of sexual samples at 0, 2, 6, 10 and 12 weeks, respectively.

**Figure 7 foods-13-00615-f007:**
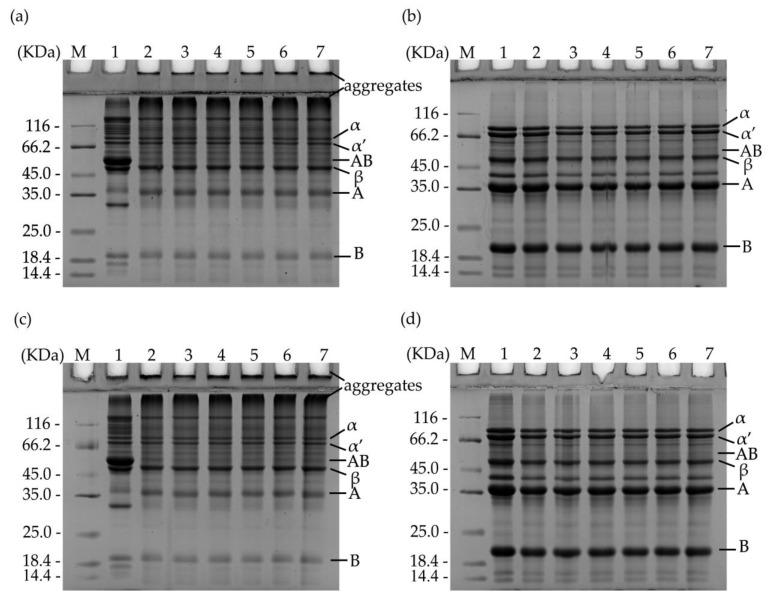
SDS-PAGE patterns of HSPI dispersions with an annealing temperature of 65 °C and different annealing times: 0 and 12 weeks. (**a**,**c**) are the non-reduced electropherograms, and (**b**,**d**) are the reduced electropherograms, respectively, where M is maker, and 1 is NSPI. Numbers 2, 3, 4, 5, 6 and 7 denote samples with annealing times of 0, 5, 10, 30, 60 and 120 min, respectively.

## Data Availability

The original contributions presented in the study are included in the article, further inquiries can be directed to the corresponding authors.

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
