# Peer review of "Improving the Storage Stability of Soy Protein Isolate through Annealing"

_foods, 2024, doi:10.3390/foods13040615_

Round 1
Reviewer 1 Report
Comments and Suggestions for Authors
The manuscript entitled: “Improving the storage stability of soy protein isolate through annealing” authored by Shenzhong Zou, Zhaojun Wang, Maomao Zeng, Zhiyong He and Jie Chen reports a new concept of food pretreatment – annealing to improve stability of soy proteins during prolonged storage. To evaluate the effects of annealing on soy protein isolates (SPI) in terms of denaturation, aggregation, and solubility of proteins the authors used appropriate and adequate methodology.
However, a few points are unclear:
1. The authors in the introduction (line 38) pointed out that keeping proteins (protein isolates, food, etc.) at low temperatures (°C) in terms of preservation of their stability generates high costs is it annealing adequate substitution since protein isolates should undergo thermal pretreatment-annealing and be stored at 37 °C for prolonged periods?
2. Moreover, one set of SPIs before annealing treatment is subjected to the temperature sterilization step. Is this step required in terms of prevention of microbial contamination due to future prolonged storage at 37 °C? What is the rationale behind different temperature treatments of SPI before annealing?
3. Currently in the literature, the annealing process is commonly associated with starch treatment. However, it is important for the authors to emphasize the limited availability of literature data on the annealing treatment of protein isolates.
4. The authors should double-check the reference [6] ( https://doi.org/10.1007/s11746-015-2684-6 ) is the reduction of solubility 63%? For which type of preparation of SPI samples?
5. Regarding reference [11] ( https://doi.org/10.1016/0022-3115(92)90533-q ) it refers to the steel grains and authors should add steel before the word grain, because considering that it is a Foods journal, readers can get the wrong impression that is related to food grains. Authors should conduct a thorough literature review and incorporate more references regarding the annealing treatment of food. This recommendation emanates from the notable distinctions between materials such as steel, metal, plastics, and food.
6. In the used formulas (2) and (3) (lines 124 and 138) authors should put a brief description of the used abbreviations.
7. Figure 1. organized in this manner is not easy to follow, the kind suggestion to the authors is to present the most representative treatment during different storage times and the rest of the results present in the Supplementary section. Also, the figure capture should be more informative with all abbreviations explained.
8. In Figure 2. and Figure 6. all the graphs should have legends. Previous suggestions for Figure 1. can also refer to Figure 3.
9. In Figure 5. in the figure capture letters denotation regarding statistical significance should be explained along with applied statistic interpretations. Are the observed differences within one group? Can be concluded that observed statistically significant differences are not among groups (0, 2, 6, 10 and 12 weeks) but it should be stated in the figure capture.
10. In Figure 7. figure captures (b) and (d) are electropherograms obtained in reducing conditions? I suppose that “NON-reduced electropherograms” is the typo?
11. In the appendix figure, the figure legend abbreviations are written in Chinese letters.
Comments on the Quality of English LanguageThe manuscript is written in good English and moderate corrections are needed.
Author Response
Dear reviewers,
We thank all the reviewers for their comments and suggestions. Please find our answers to the reviewer’s comments below:
-Reviewer 1
- The authors in the introduction (line 38) pointed out that keeping proteins (protein isolates, food, etc.) at low temperatures (°C) in terms of preservation of their stability generates high costs. is it annealing adequate substitution since protein isolates should undergo thermal pretreatment-annealing and be stored at 37 °C for prolonged periods?
Thank you for your question. The results of this study show that while annealing proves effective in mitigating the rate of solubility decrease in highly denatured proteins, it does not suffice to entirely replace the impact of cryogenic storage.
- Moreover, one set of SPIs before annealing treatment is subjected to the temperature sterilization step. Is this step required in terms of prevention of microbial contamination due to future prolonged storage at 37 °C? What is the rationale behind different temperature treatments of SPI before annealing?
In this study, we chose this condition with reference to the operation of ultra-high temperature sterilization of soy protein in industrial production; however, the UHT step is not only a microbiological issue (sterilization), but also a step to control the aggregate size and to modify soy protein. Therefore, the main purpose of different temperature treatments before annealing was to create SPIs with different degrees of denaturation to observe the effect of annealing on the storage stability of SPIs with different degrees of aggregation.
- Currently in the literature, the annealing process is commonly associated with starch treatment. However, it is important for the authors to emphasize the limited availability of literature data on the annealing treatment of protein isolates.
Thank you for your suggestion. We have emphasized the limited availability of literature data on the annealing treatment of protein isolates in line 58 as follows:
“However, little has been reported on the annealing treatment of soybean protein isolate.”
- The authors should double-check the reference [6] ( https://doi.org/10.1007/s11746-015-2684-6 ) is the reduction of solubility 63%? For which type of preparation of SPI samples?
We have checked the reference [6] ( https://doi.org/10.1007/s11746-015-2684-6 ) is the reduction of solubility 63%, but it is stored in an aqueous solution. We have changed the formulation of this sentence in the article as follows:
“For instance, the solubility of SPI in water (2%) decreased by 63% after one year of storage at 42°C.”
- Regarding reference [11] ( https://doi.org/10.1016/0022-3115(92)90533-q ) it refers to the steel grains and authors should add steel before the word grain, because considering that it is a Foods journal, readers can get the wrong impression that is related to food grains. Authors should conduct a thorough literature review and incorporate more references regarding the annealing treatment of food. This recommendation emanates from the notable distinctions between materials such as steel, metal, plastics, and food.
Thank you for your suggestion. We have added the term “steel” before the word “grain” to avoid any confusion regarding the nature of the material. In addressing the concern about the literature review, we acknowledge the importance of emphasizing the relevance of annealing treatment specifically in the context of food-related materials. While reference [11] initially discusses steel grains, our intention was to establish a broader understanding of the annealing process by highlighting its successful applications in various fields, including metal and polymer materials. By drawing parallels with soybean protein isolate, which falls within the category of polymers, we aimed to showcase the applicability of annealing principles across diverse materials. Despite the distinct composition and structure of soybean protein isolate compared to steel grains, the fundamental principles of annealing remain similar. This provides a feasibility analysis of the annealing of soybean protein isolate and it is an important innovation of this study.
- In the used formulas (2) and (3) (lines 124 and 138) authors should put a brief description of the used abbreviations.
Thank you for your suggestion. We have put a brief description of the used formulas (2) and (3) (lines 124 and 138) as follows:
“Incubation at room temperature in the dark for 15 min was followed by absorbance measurement at 412 nm to calculate total free sulfhydryl groups using the following formula:
SH denotes sulfhydryl group, A412 denotes the absorbance of the sample at 412 wavelength, and C denotes the concentration of protein.”
“Absorbance was measured at 412 nm. Disulfide bonds content was calculated using the following formula:
-S-S- denotes disulfide bond, SHT denotes total sulfhydryl and SHF denotes total free sulfhydryl.”
- Figure 1. organized in this manner is not easy to follow, the kind suggestion to the authors is to present the most representative treatment during different storage times and the rest of the results present in the Supplementary section. Also, the figure capture should be more informative with all abbreviations explained.
Thank you for your kind suggestion. We have chosen the most representative treatment to present and and the rest of the results present in the Supplementary section of Figure 1. The results are as follows:
Figure 1. The solubility of soy protein isolate with or without annealing treatment as a function of storage durations. NSPI: native soy isolate protein; HSPI: denatured soy isolate protein; H-65-5 denotes that the high denatured soy protein isolate was subjected to annealing treatment at 65°C for 5 min; H-65-30 denotes highly denatured soybean isolate annealed at 65°C for 30 minutes; others by analogy.
Appendix
|
|
|
|
|
Figure A1. The solubility of soy protein isolate with or without annealing treatment as a function of storage durations. NSPI: native soy isolate protein; LSPI: low denatured soy isolate protein and HSPI: denatured soy isolate protein; L-45-5 denotes that the low denatured soy protein isolate was subjected to annealing treatment at 45°C for 5 min, others by analogy.
- In Figure 2. and Figure 6. all the graphs should have legends. Previous suggestions for Figure 1. can also refer to Figure 3.
Thank you for your kind suggestion. We've added legends to Figures 2 and 3, as well as other Figures. Meanwhile, we have chosen the most representative treatment to present and and the rest of the results present in the Supplementary section of Figure 3. The results are as follows:
Figure 3. Molecular weight distribution of HSPI samples with annealing temperature of 65°C, different annealing times, and different storage times; (a, a1) (b, b1) indicate the molecular weight distribution of the samples at 0 and 10 weeks, respectively.
Figure 4. Sulfhydryl group content of HSPI samples with different annealing temperatures of 65°C, different annealing times, and different storage times; (a) (b) (c) (d) indicate the changes of surface sulfhydryl groups, total sulfhydryl groups, total free sulfhydryl groups, and disulfide bonds of the samples during the storage, respectively.
|
|
Figure 6. Intrinsic emission fluorescence spectra of HSPI samples with annealing temperature of 65°C, different annealing times, and different storage times. (a) (b) (c) (d) (E) Intrinsic emission fluorescence spectra of sexual samples at 0, 2, 6, 10 and 12 weeks, respectively.
Appendix
Figure A2. Molecular weight distribution of HSPI samples with annealing temperature of 65°C, different annealing times, and different storage times; (c, c1) (d, d1) (e, e) indicate the molecular weight distribution of the samples at 2, 6 and 12 weeks, respectively.
- In Figure 5. in the figure capture letters denotation regarding statistical significance should be explained along with applied statistic interpretations. Are the observed differences within one group? Can be concluded that observed statistically significant differences are not among groups (0, 2, 6, 10 and 12 weeks) but it should be stated in the figure capture.
Thank you for your kind suggestion. We have explained that the observed differences are within a group, and we have also indicated the corresponding statistical significance with a letter. The results are as follows:
Figure 5. Surface hydrophobicity of HSPI samples at annealing temperature of 65°C, different annealing times, and different storage times. Different letters indicate significant differences between different samples between different samples in the same week(P< 0.05).
- In Figure 7. figure captures (b) and (d) are electropherograms obtained in reducing conditions? I suppose that “NON-reduced electropherograms” is the typo?
Thank you for your kind suggestion. We have made changes here, and Figures (b) and (d) were indeed electrograms obtained under reducing conditions. The results are as follows:
Figure 7. SDS-PAGE patterns of HSPI dispersions with annealing temperature of 65°C and different annealing times, 0 and 12 weeks. (a) (c) are the non-reduced electropherograms, (b) (d) are the reduced electropherograms respectively; where M is maker, 1 is NSPI, 2, 3 ,4, 5, 6, 7 denote samples with annealing times of 0, 5, 10, 30, 60 and 120 min, respectively.
- In the appendix figure, the figure legend abbreviations are written in Chinese letters.
Thank you for your suggestion. We have fixed the error here and the result is as follows:
Figure A3. It shows the relative density of aggregates in the unreduced SDS-PAGE mode.

Reviewer 2 Report
Comments and Suggestions for Authors
Review the citation order. There seems to be a gap between the Introduction section that ends with [16] and the Material and Methods, 2.2, in which [35] appears.
In the Introduction section, the authors mention previously used methods to limit the negative influence of different factors on soy protein stability. Among them are also the antioxidants that did not give satisfactory results. In Conclusion, the possible use of annealing and chemical methods is suggested as synergism. What are the authors based on when they make this statement? Please give support references for the idea.
The paper could be accepted for publication after minor changes. It has to be revised by the authors and resubmitted with suggested modifications specified in the reviewer’s comments.
Comments on the Quality of English LanguageThe English-style overview is recommended due to the presence of the same word in a phrase. Please use synonyms or paraphrases.
Author Response
Dear reviewers,
We thank all the reviewers for their comments and suggestions. Please find our answers to the reviewer’s comments below:
Reviewer 2
- Review the citation order. There seems to be a gap between the Introduction section that ends with [16] and the Material and Methods, 2.2, in which [35] appears.
Thank you for your kind suggestion. We have changed the incorrect citation order and checked again the entire manuscript.
- In the Introduction section, the authors mention previously used methods to limit the negative influence of different factors on soy protein stability. Among them are also the antioxidants that did not give satisfactory results. In Conclusion, the possible use of annealing and chemical methods is suggested as synergism. What are the authors based on when they make this statement? Please give support references for the idea.
Thanks for your question. The addition of antioxidants been explored as a means to preserve the storage stability of soybean isolates, even though the outcomes have not been notably favourable. In our study, we found that annealing also significantly contributes to the preservation of soy protein stability during storage. Our suggestion in the Conclusion section to explore the synergistic use of annealing and chemical methods is based on the observed benefits of both techniques in enhancing stability.